# Mental Health Is a Family Affair—Systematic Review and Meta-Analysis on the Associations between Mental Health Problems in Parents and Children during the COVID-19 Pandemic

**DOI:** 10.3390/ijerph20054485

**Published:** 2023-03-02

**Authors:** Markus Stracke, Miriam Heinzl, Anne Dorothee Müller, Kristin Gilbert, Anne Amalie Elgaard Thorup, Jean Lillian Paul, Hanna Christiansen

**Affiliations:** 1Department of Clinical Child and Adolescent Psychology, Philipps University Marburg, 35032 Marburg, Germany; 2Research Unit, Child and Adolescent Mental Health Center, 2100 Copenhagen, Denmark; 3Institute for Clinical Medicine, University of Copenhagen, 2200 Copenhagen, Denmark; 4Mental Health Research Program, The Village, Ludwig Boltzmann Gesellschaft, 6020 Innsbruck, Austria; 5Division of Psychiatry I, Department of Psychiatry, Psychotherapy and Psychosomatics, Medical University Innsbruck, 6020 Innsbruck, Austria

**Keywords:** parents with mental illness, parental mental illness, children of parents with mental illness, transgenerational transmission, family health, parenting stress, pandemic, COVID-19

## Abstract

As a multidimensional and universal stressor, the COVID-19 pandemic negatively affected the mental health of children, adolescents, and adults worldwide. In particular, families faced numerous restrictions and challenges. From the literature, it is well known that parental mental health problems and child mental health outcomes are associated. Hence, this review aims to summarize the current research on the associations of parental mental health symptoms and child mental health outcomes during the COVID-19 pandemic. We conducted a systematic literature search in Web of Science (all databases) and identified 431 records, of which 83 articles with data of over 80,000 families were included in 38 meta-analyses. A total of 25 meta-analyses resulted in significant small to medium associations between parental mental health symptoms and child mental health outcomes (*r* = 0.19 to 0.46, *p* < 0.05). The largest effects were observed for the associations of parenting stress and child mental health outcomes. A dysfunctional parent–child interaction has been identified as a key mechanism for the transmission of mental disorders. Thus, specific parenting interventions are needed to foster healthy parent–child interactions, to promote the mental health of families, and to reduce the negative impacts of the COVID-19 pandemic.

## 1. Introduction

Worldwide, the lives of children, adolescents, and adults have been impacted by the COVID-19 pandemic over the past two years [1]. To contain the spread of the SARS-CoV-2 virus and to prevent the collapse of the healthcare system, governments around the globe have imposed numerous public health measures and (lockdown) restrictions [2], thus radically changing almost all areas of life for almost everybody [1,3]. 

Characterized by (1) its global character and unclear duration, (2) its impact on multiple areas of life, (3) experiencing feelings of loss of control and helplessness, (4) its systemic impact on society, and (5) limited or blocked access to protective factors, the COVID-19 pandemic has been described as a multidimensional and universal stressor [3,4,5]. According to the transactional stress model [6], stressful situations lead to negative consequences (such as mental health problems) if the resources of an individual are not sufficient to react to the stressor. According to vulnerability stress models, e.g., [7], the development of psychopathology depends on the interaction of stressful events and pre-existing vulnerabilities in balance with protecting factors, buffering systems, and resilience. Thus, it has been argued that during the COVID-19 pandemic, especially vulnerable groups, such as people with pre-existing mental illnesses, could be susceptible to the negative impacts of the pandemic on their mental health [4,5,8]. In line with this, studies revealed that pre-existing mental health problems in adults [9,10], as well as in children and adolescents [11], are a significant risk factor for mental health problems during the COVID-19 pandemic. In a cohort study with individuals with pre-existing mental illness, 60% of the participants reported that their mental health had worsened during the pandemic [12]. Further, in a continuously growing number of studies, the negative effects of the COVID-19 pandemic on the mental health of adults—for systematic reviews and meta-analyses, see [13,14,15,16]—as well as children and adolescents—for systematic reviews and meta-analyses, see [11,17,18,19,20]—are now well documented. For adults, meta-analyses have revealed heightened prevalence rates of anxiety (15.15% to 32.6%) and depression (15.97% to 31.4%) [13,15,21], as well as heightened prevalence rates of anxiety (25.2% to 34.5%) and depression (20.5% to 41.7%) for children and adolescents [19,20], in the course of the pandemic.

During the COVID-19 pandemic, families, in particular, faced numerous challenges. Due to the closure of schools and childcare facilities, leisure time activities, and places providing social support, the routines of families were disrupted. From one day to the next, families were required to adjust to quarantine and social distancing and to re-organize their everyday lives, thus potentially stressing all family members [1]. Parents had to manage all childcare as well as schooling demands, while still meeting their other requirements (job, household, etc.). In comparison to adults without caregiving responsibilities, higher rates of mental health problems were reported for parents and caregivers during the COVID-19 pandemic [22]. 

Whereas we refer in this article to the experience of stress in different life domains (e.g., daily routines, household, work, financial, health) as *general* stress, *parenting* stress is understood as a specific form of stress, experienced only by parents. Parenting stress relates specifically to the demands of being a parent and the interaction with one or more children [23]. In line with the transactional stress model [6], parenting stress occurs when the experienced demands of being a parent exceed the perceived resources as a parent. Parenting stress has been frequently associated with negative child outcomes, such as internalizing and externalizing symptoms, as well as children’s depression and anxiety (*r* = 0.15 to 0.37) [24,25,26], and was found to be higher during the COVID-19 pandemic [27,28].

From the literature, it is also well known that parental mental health problems and child mental health outcomes are associated [29,30,31]. Meta-analyses revealed small associations between children’s inter- and externalizing symptoms and maternal and paternal psychopathology (*r* = 0.14 to 0.18) [30], as well as maternal and paternal depression (*r* = 0.19 to 0.24) [29,30,31]. Furthermore, children of parents with a mental illness (COPMI) have a higher risk of developing a mental disorder themselves [32,33,34]. For COPMI, a lifetime risk of developing a serious mental disorder between 41% and 77%, with subclinical symptoms occurring more often and earlier in life, is reported [35].

In an integrated model explaining how the COVID-19 pandemic has affected families, Prime et al. [36] postulated that social disruptions due to the pandemic led to higher levels of psychological distress in caregivers, which, moderated by changes in the relationship quality of all family members, affected child adjustment, especially in families with pre-existing vulnerabilities. Consistently, in a pre-COVID-19 sample, positive associations between parents’ general stress and children’s internalizing and externalizing symptoms were found that were partially mediated by family conflict [37].

Taken together, families with pre-existing mental illness, and specifically children of parents with a mental illness, constitute an especially vulnerable group to the potential negative consequences of the COVID-19 pandemic. As specific studies on COPMI during the COVID-19 pandemic are sparse, e.g., [38], we aimed to address this gap by systematically investigating the association of parental and child mental health problems during the COVID-19 pandemic.

The current study summarizes the available research on the associations of parental mental health symptoms (psychopathology, depression, anxiety, general stress, parenting stress) and child mental health outcomes (psychopathology, internalizing symptoms, depression, anxiety, externalizing symptoms, general stress) during the COVID-19 pandemic. Our research questions are as follows:(1)Are parental mental health problems prior to the COVID-19 pandemic associated with child mental health outcomes during the COVID-19 pandemic?(2)Are current parental mental health symptoms associated with child mental health outcomes during the COVID-19 pandemic?(3)Are parents’ general stress levels associated with child mental health outcomes during the COVID-19 pandemic?(4)Is parenting stress associated with child mental health outcomes during the COVID-19 pandemic?

## 2. Methods

To answer our research questions, we conducted a systematic review [39]. The review followed the preferred reporting items for systematic reviews and meta-analyses (PRISMA) guidelines [40] and was registered in PROSPERO, registration number CRD42021273376.

### 2.1. Literature Search

We conducted a systematic literature search in Web of Science (all databases) on 25 May 2022. The search terms included terms on parental mental illness or stress, the COVID-19 pandemic, and child mental health outcomes. The full search strategy is provided in the Appendix A. The search was limited to articles published between 2020 and 25 May 2022. Original studies reporting on the association of parental mental health problems or stress and child mental health outcomes during the COVID-19 pandemic were included. No further in- or exclusion criteria were applied.

### 2.2. Study Selection

All abstracts were screened by two authors separately (M.S. and M.H.). Disagreements were resolved by discussion and by involving a third author (H.C.) until a consensus was reached. The full texts of the remaining articles were assessed for eligibility by two authors separately (M.S. and M.H.). Again, disagreements were resolved by discussion and by involving a third author (H.C.) until a consensus was reached.

### 2.3. Data Extraction

In order to address the research questions, the following information was extracted from the included studies: author(s), year of publication, title, country, study design (e.g., online survey, cross-sectional), parent/child report, study period, study participants (sample size, gender distribution, children’s age), measurement of parental mental health problems, measurement of parents’ general stress, measurement of parenting stress, measurement of children’s mental health outcomes, association of parental mental health problems/stress and children’s mental health outcomes. The data were either extracted by M.S. or M.H. and, afterwards, double-checked by the other (M.S. or M.H.). For meta-analytic calculations, Pearson’s product–moment correlation *r* was used. If results from more than one measurement point were reported in a study, data on the first COVID-19 measurement point were extracted. If studies reported only adjusted correlations or results from (multivariate or hierarchical) regression analyses, the corresponding authors were contacted and asked to provide Pearson’s product–moment correlations *r*. In case of Spearman’s rho correlations, these were transformed into a Pearson’s product–moment correlation using the formula by Rupinski and Dunlap [41]. If studies reported more than one correlation for a domain of interest, the correlations were extracted separately and later combined into a single correlation. For this, the correlations were transformed into Fisher’s z scores and averaged (arithmetic mean). Afterwards, the average was inversely z-transformed back into a Pearson’s product–moment correlation *r*. This procedure was applied to combine (1) correlations that were reported on the item level into internalizing/externalizing symptom scales, (2) correlations reported for subscales of different internalizing/externalizing symptoms into internalizing/externalizing symptom scales, (3) correlations for internalizing and externalizing symptoms into one general psychopathology scale (for children), and (4) correlations for different symptoms into one general psychopathology scale (for adults).

### 2.4. Risk of Bias

Two authors (M.S. and M.H.) evaluated the quality and risk of bias of the included studies using the AXIS tool [42], which is a standardized and robust critical appraisal tool to assess the quality and risk of bias of cross-sectional studies. Since only cross-sectional data were extracted from longitudinal studies, we also used the AXIS tool for the quality assessment of the longitudinal studies. Again, disagreements were resolved by discussion and by involving a third author (H.C.) until a consensus was reached. The information from the AXIS tool was condensed into a final risk of bias rating for each study: low (18–20), medium (14–17), or high (0–13) risk of bias [43].

### 2.5. Systematic Synthesis and Statistical Analysis

To synthesize the results of the included studies, we structured them according to the types of parental and child mental health problems. Studies reporting on more than one parental mental health problem or child mental health outcome were included in each respective section. Since we included many studies that reported a range of different effect sizes, we decided not to present each study more comprehensively, but to statistically summarize the available evidence [44]. Thus, wherever possible, meta-analyses, which are considered to be the highest level of evidence [45], were computed to combine the different effect sizes of the included studies. Meta-analyses were calculated with the free online meta-analysis tool Meta-Mar [46,47]. As proposed for small numbers of studies and heterogeneity, the random effect model based on the inverse variance approach with Knapp–Hartung adjustment [48,49] was used. No covariates were included in the model. Correlations were Fisher’s z-transformed, weighted by sample size, and combined. The overall Fisher’s z score was afterwards inversely z-transformed back into a Pearson’s product–moment correlation *r*. Overall, effects were interpreted according to Cohen [50]. Heterogeneity was tested using the *Q* test, the *T*^2^ statistic, and the *I*^2^ statistic and interpreted according to Higgins et al. [51]. *Fail-safe N* was calculated for all significant meta-analyses according to Rosenthal’s approach [52].

## 3. Results

The literature search yielded overall 431 records. After the removal of duplicates, 430 abstracts were screened, and 285 records were excluded. The full texts of the remaining 145 articles were screened for eligibility. Finally, a total of 83 articles were allocated to the four research questions and included in the analyses. See Figure 1 for the PRISMA flow chart [40].

Data on 86,658 parents and more than 82,312 children were examined in this review. The median number of parents per study was 306.5 (range: 21 to 29,202) and of children was 297 (range: 26 to 29,202). The included studies were mainly from North America (32 studies; 27 of which were from the USA), Europe (31 studies; 14 of which were from Italy), and Asia (15 studies; five of which were from China). Fifty studies were designed as cross-sectional, whereas 33 studies reported longitudinal data. The included studies were conducted between January 2020 and April 2021, but most studies reported data that were collected in April or May 2020 (42 studies each). The age range of included children and adolescents was between 0 and 21 years (*M* = 9.18). Since children’s mean age was not reported in all studies, we also coded age categories (children < 12 years, adolescent ≥ 12 years). A total of 39 studies focused on children, 15 studies on adolescents, and 29 studies reported data on both children and adolescents. In 59 studies, child/adolescent outcomes were reported only by parents, in two studies only by adolescents, and in 22 studies by both parents and children/adolescents. A detailed description of the study characteristics of all included studies can be found in the Appendix A.

Reported parental mental health problems were categorized into prior parental psychopathology (four studies), prior parental depressive symptoms (five studies), current parental psychopathology (34 studies), current parental depressive symptoms (31 studies), current parental anxiety symptoms (25 studies), parents’ current general stress (39 studies), and current parenting stress (18 studies). Reported child mental health outcomes were categorized into children’s psychopathology (45 studies), children’s internalizing symptoms (42 studies), children’s depressive symptoms (14 studies), children’s anxiety symptoms (19 studies), children’s externalizing symptoms (34 studies), and children’s current general stress (13 studies).

The most commonly used measures of parental mental health symptoms were the Generalized Anxiety Disorder Scale (12 studies), the Patient Health Questionnaire (11 studies), and the Depression–Anxiety–Stress Scale (nine studies). Parents’ current general stress was mostly assessed with ad hoc developed items (14 studies) or the Perceived Stress Scale (nine studies). Parenting stress was mostly assessed with the Parental Stress Scale and the Parenting Stress Index (each seven studies). The most commonly used measures for child mental health symptoms were the Strengths and Difficulties Questionnaire (27 studies) and the Child Behavior Checklist (14 studies). Children’s current general stress was mostly assessed with the Perceived Stress Scale (five studies) or with ad hoc developed items (four studies) (for details, see Appendix A).

Most studies were rated as having a medium risk of bias (69 studies) according to the AXIS tool (see Appendix A). The most commonly raised concerns were that (1) some of the risk factors and outcome variables were not measured with trialed, piloted, or previously published instruments (58 studies); (2) the sample size was not justified (65 studies); and (3) concerns about a possible non-response bias were raised. This was either because no information about non-responders was given (71 studies), no measures were undertaken to address and categorize non-responders (69 studies), or because no response rate was reported or the response rate raised concerns about non-response bias (62 studies).

To avoid redundancy, the results are presented summarized for each research question. Details on all observed overall effects are displayed in Table 1. A summary of the observed significant effects can be found in Table 2.

### 3.1. Research Question 1: Are Parental Mental Health Problems Prior to the COVID-19 Pandemic Associated with Child Mental Health Outcomes during the COVID-19 Pandemic?

The association of parental mental health problems prior to the COVID-19 pandemic and child mental health outcomes during the COVID-19 pandemic was investigated in eight studies [53,54,55,56,57,58,59,60]. Nine meta-analyses were conducted that resulted in overall small effects (*r* = 0.09 to 0.29, *p* = 0.01 to 0.30) with low to high heterogeneity. Due to the small number of included studies in the meta-analyses concerning this research question (*k* ≤ 4 for all meta-analyses), only the meta-analysis on the association between prior parental depressive symptoms and current children’s internalizing symptoms reached statistical significance (*r* = 0.19, *p* = 0.01). For this meta-analysis, no heterogeneity was observed (*Q* = 0.27, *p* = 0.87, *I*^2^ = 0.0%).

### 3.2. Research Question 2: Are Current Parental Mental Health Symptoms Associated with Child Mental Health Outcomes during the COVID-19 Pandemic?

A possible association between parental mental health symptoms and child outcomes during the COVID-19 pandemic was investigated in 54 studies [28,53,54,55,56,58,59,60,61,62,63,64,65,66,67,68,69,70,71,72,73,74,75,76,77,78,79,80,81,82,83,84,85,86,87,88,89,90,91,92,93,94,95,96,97,98,99,100,101,102,103,104,105,106]. Eighteen meta-analyses resulted in overall small to medium effects (*r* = 0.22 to 0.39, *p* < 0.001 to *p* = 0.20) with significant medium to high heterogeneity (*Q* = 14.33 to 226.39, *p* < 0.05, *I*^2^ = 46.1 to 97.2%). Again, due to the small number of included studies in some of the meta-analyses, three meta-analyses (parental psychopathology/parental depressive symptoms/parental anxiety symptoms and children’s general stress; all *k* ≤ 5) did not reach statistical significance.

### 3.3. Research Question 3: Are Parents’ General Stress Levels Associated with Child Mental Health Outcomes during the COVID-19 Pandemic?

The association between parents’ general stress and child outcomes during the COVID-19 pandemic was investigated in 39 studies [54,58,60,61,66,68,72,74,76,78,79,83,90,91,93,94,95,98,107,108,109,110,111,112,113,114,115,116,117,118,119,120,121,122,123,124,125,126,127]. Six meta-analyses were conducted that revealed statistically significant overall small to medium effects (*r* = 0.29 to 0.34, *p* < 0.001). For all these meta-analyses, significantly high heterogeneity was observed (*Q* = 29.68 to 433.74, *p* < 0.001, *I*^2^ = 76.4 to 95.6%).

### 3.4. Research Question 4: Is Parenting Stress Associated with Child Mental Health Outcomes during the COVID-19 Pandemic?

The association between parenting stress and child outcomes during the COVID-19 pandemic was investigated in 18 studies [28,55,64,66,82,85,95,96,107,122,123,128,129,130,131,132,133,134]. Five meta-analyses revealed overall medium to high effects (*r* = 0.36 to 0.51, *p* < 0.001 to *p* = 0.28) with medium to high heterogeneity. Again, due to the small number of included studies (*k* = 2), two meta-analyses (parenting stress and children’s depressive symptoms/children’s general stress) did not reach statistical significance. For the three meta-analyses that reached statistical significance, significant medium to high heterogeneity was observed (*Q* = 15.83 to 95.89, *p* < 0.05, *I*^2^ = 55.8 to 93.7%).

## 4. Discussion

As a multidimensional and universal stressor [3,4,5], the COVID-19 pandemic has had and still has negative effects on the mental health of children, adolescents, and adults worldwide [11,13,14,15,16,17,18,19,20]. Pre-existing mental health problems were found to be a significant risk factor for mental health problems during the COVID-19 pandemic [9,11]. In particular, families were faced with numerous challenges due to the imposed public health measures and (lockdown) restrictions. From the literature, it is well known that parental mental health problems and child mental health outcomes are associated [29,30,31], and that COPMI have a higher risk of developing a mental disorder themselves [32,33,34]. Thus, it was hypothesized that COPMI would be especially susceptible to the negative effects of the COVID-19 pandemic. In light of this, specific studies concerning COPMI during the COVID-19 pandemic were found to be very sparse, e.g., [38]. The aim of the current study was thus to address this research gap by systematically summarizing the current research on the associations between parental mental health symptoms and child mental health outcomes during the COVID-19 pandemic.

With respect to research question 1 on the association of parental mental health problems prior to the COVID-19 pandemic and child mental health outcomes during the COVID-19 pandemic, we found a small but significant association between parental depression prior to the COVID-19 pandemic and children’s internalizing symptoms during the COVID-19 pandemic (*r* = 0.19). This finding is in line with pre-pandemic data, which also found small, but significant, associations between maternal (*r* = 0.23) [29] as well as paternal depression (*r* = 0.24) [31] and child internalizing symptoms. Within the first research question, eight more meta-analyses were calculated, which resulted in overall small effects. Due to the small number of included studies (*k* ≤ 4), these meta-analyses did not reach statistical significance.

Regarding the association of current parental mental health symptoms and child mental health outcomes during the COVID-19 pandemic (research question 2), small to medium effects were observed. Parental psychopathology (*r* = 0.27 to 0.36), parental depressive symptoms (*r* = 0.25 to 0.39), and parental anxiety symptoms (*r* = 0.22 to 0.31) correlated significantly with children’s mental health outcomes (psychopathology, internalizing symptoms, depressive symptoms, anxiety symptoms, and externalizing symptoms) during the COVID-19 pandemic. Although of larger magnitude, our findings are again in line with pre-pandemic data that found small effects for the association of parental psychopathology, as well as parental depression, and children’s internalizing and externalizing symptoms and general psychopathology (*r* = 0.14 to 0.24) [29,30,31]. Within the second research question, three more meta-analyses were calculated, which resulted in overall medium effects. Due to the small number of included studies (*k* ≤ 5), these meta-analyses did not reach statistical significance, though.

For research question 3 on the association of parents’ general stress and child mental health outcomes (psychopathology, internalizing symptoms, depressive symptoms, anxiety symptoms, and externalizing symptoms) during the COVID-19 pandemic, significant small to medium effects were observed (*r* = 0.29 to 0.34). This finding is also in line with pre-pandemic data that report similar correlations of medium magnitude (*r* = 0.35) for the association between parents’ general stress and children’s internalizing and externalizing symptoms [37].

Regarding the association of parenting stress with child mental health outcomes during the COVID-19 pandemic (research question 4), significant medium effects were observed. Parenting stress correlated significantly with children’s psychopathology as well as children’s internalizing and externalizing symptoms (*r* = 0.44 to 0.46). Although of larger magnitude, our findings are again in line with pre-pandemic research, where small to medium associations (*r* = 0.15 to 0.31) between parenting stress and children’s internalizing and externalizing symptoms were reported [24,25]. Within research question 4, two more meta-analyses were calculated, which, again due to the small number of included numbers (*k* = 2), did not reach statistical significance.

Taken together, we were able to replicate pre-pandemic findings that parental mental health problems are associated with children’s mental health outcomes [24,25,26,29,30,31,37] and to show that the COVID-19 pandemic, as a multidimensional and universal stressor, intensifies pre-existing and already known risk mechanisms, such as the transgenerational transmission of mental disorders (TTMD) [35]. Thereby, we can add to the strong, empirical evidence that not only parental depression but any parental mental health problem should be seen and understood as a family affair [135]. While it is clear from the literature that COPMI can be seen as the next generation of people with a mental illness [32], they are often overseen by policy makers, researchers, and the society at large, resulting in invisibility, e.g., [136]. The low number of specific studies concerning COPMI and family mental health indicates that COPMI were also forgotten at large during the COVID-19 pandemic, and their heightened needs in view of a long-lasting, multidimensional stressor were severely neglected by societies and politics [3].

In our meta-analyses, the largest effects were found for the association of parenting stress and child mental health outcomes. This finding fits well with existing theoretical models concerning the TTMD [35], as well as the conceptual framework of family well-being during the COVID-19 pandemic [36]. The TTMD identifies the parent–child interaction as a key mechanism for the transmission of mental disorders [35]. Consistently, Prime et al. [36] postulated that the association between heightened caregiver distress (due to the COVID-19 pandemic) and child outcomes is moderated by the parent–child relationship quality. In line with these models, Chung et al. [137] showed that the negative impact of the COVID-19 pandemic increased parenting stress and in turn reduced parent–child relationship quality. Similarly, Spinelli et al. [122] found that the association of the negative impact of the COVID-19 pandemic on children’s symptomatology was mediated by parents’ general stress as well as parenting stress. Babore et al. [28] also reported that the association between parents’ psychopathology and children’s symptomatology was mediated by parenting stress. Furthermore, Cohodes et al. [112] demonstrated that parents who effectively dealt with children’s negative emotions and maintained stable home routines were able to buffer the negative impact of the COVID-19 pandemic on children’s symptomatology. On the other hand, if parenting stress and parental anxiety were reported to be high, the COVID-19 pandemic had a more negative impact on children’s symptomatology [112].

Given that the pre-pandemic literature has shown that parental psychopathology and parenting stress are related [138] and that parent–child interactions, as well as parenting skills, are impaired for parents with a mental illness compared to parents without a mental illness [139,140], it is likely that parents with a mental illness were especially stressed by the COVID-19 pandemic and, hence, families with a parent with a mental illness were less able to buffer the negative impact of the COVID-19 pandemic on their children. Thus, the lockdown restrictions, which led to the closure of schools, childcare facilities, and leisure time activities, might have been and might still be especially detrimental for COPMI, since those external buffering systems were also no longer available.

### 4.1. Limitations

In light of our research findings, several limitations have to be considered. First, most of the studies included in our meta-analyses report cross-sectional data. Thus, causality is not clear. Although we argue that parental mental health problems impact child and adolescent mental health, this relationship could also be the other way around, with child health affecting parental mental health as well [36,53,65]. Second, most child and adolescent mental health outcomes were reported by the parents. Thus, the report might be biased by their perceptions and their own psychopathology or mental health status [141]. Third, heterogeneity was found to be medium or high for all significant meta-analyses (except for the one regarding research question 1). Due to the low number of included studies in many meta-analyses, we were not able to conduct moderator analyses, but further research should investigate moderators such as age [124], sex, lockdown duration [11], and months into the pandemic [91]. Fourth, most studies were rated as having a medium risk of bias. Thus, results might be biased, and more high-quality longitudinal studies are needed to draw definite conclusions.

### 4.2. Implications for Research and Practice

Our findings have several implications for further research as well as for clinical practice. To better understand the role of the parent–child interaction in the TTMD, subsequent studies should further investigate the association between parenting stress and parental as well as children’s psychopathology. A better understanding of this association will help in creating better-tailored interventions for COPMI. Since mental health should always be understood as a family affair, clinical interventions to reduce the negative impact of the COVID-19 pandemic on the mental health of children, adolescents, and parents should target three different levels: the children and adolescents, the parents, and the whole family. On the one hand, specific interventions, e.g., with methods from cognitive–behavioral therapy as the current treatment of choice for mental disorders in general [142], are needed to improve the mental health of children, adolescents, and parents. On the other hand, interventions need to target the whole family. The enhancement of parenting skills has been found to be a significant mediator in improving child outcomes [143] and is, therefore, already an important aspect of prevention programs for COPMI [144,145,146]. During the COVID-19 pandemic, international initiatives [147] or low-threshold interventions such as “Families Under Pressure” [148,149] also aimed to enhance parenting skills, to promote healthy parent–child interactions, and, thereby, to buffer the negative impact of the COVID-19 pandemic on children’s symptomatology.

## 5. Conclusions

To our knowledge, this is the first study summarizing the current research on the association of parental mental health symptoms and child mental health outcomes during the COVID-19 pandemic. In line with pre-pandemic studies, we found significant associations between parental mental health problems or stress and child mental health outcomes. Our findings demonstrate that it is of the utmost importance to not assess the negative impact of the COVID-19 pandemic on the mental health of children, adolescents, and adults separately, since the mental health problems of family members are interconnected and likely to affect each other. Since the largest effects were found for the association of parenting stress and child mental health outcomes, besides separate interventions promoting the mental health of children, adolescents, and parents, also specific parenting interventions are needed to foster family health and parent–child interactions, to promote the mental health of families, and to reduce the negative impacts of the COVID-19 pandemic.

## Figures and Tables

**Figure 1 ijerph-20-04485-f001:**
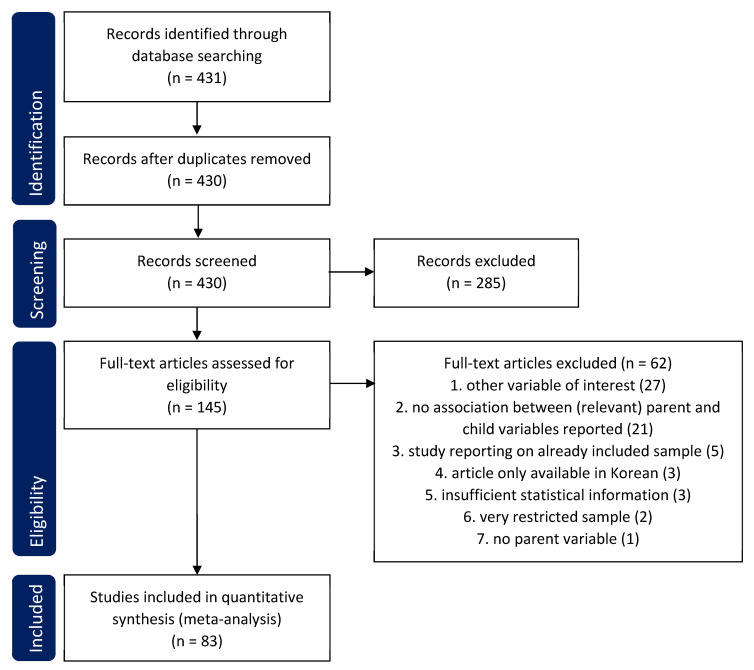
PRISMA flow diagram.

**Table 1 ijerph-20-04485-t001:** Summary of meta-analytic results on associations between parental variables prior to and during the COVID-19 pandemic and children’s mental health outcomes during the COVID-19 pandemic.

Research Question	Parental Variable	Child Outcome	Meta-Analytic Results									
			*k*	*o*	*r*	95% CI	*z*	*p*-Value	*Q*	*I* ^2^	*T* ^2^	*Fail-Safe N*
1. Are parental mental health problems prior to the COVID-19 pandemic associated with child mental health outcomes during the COVID-19 pandemic?	Psy-pathol	Psy-pathol	2	351	0.23	(−0.85; 0.94)	1.96	0.30	4.65, *p* = 0.03	78.5%	0.022	
	Psy-pathol	Int symp	4	754	0.18	(−0.07; 0.40)	2.28	0.11	13.30, *p* < 0.01	77.4%	0.019	
	Psy-pathol	Dep symp	2	403	0.16	(−0.29; 0.56)	4.53	0.14	0.53, *p* = 0.47	0.0%	0	
	Psy-pathol	Anx symp	2	403	0.09	(−0.03; 0.21)	9.61	0.07	0.04, *p* = 0.85	0.0%	0	
	Psy-pathol	Ext symp	2	351	0.23	(−0.49; 0.76)	3.87	0.16	1.25, *p* = 0.26	20.2%	0.002	
	Dep symp	Psy-pathol	2	188	0.29	(−0.73; 0.91)	3.04	0.20	1.65, *p* = 0.20	39.5%	0.007	
	Dep symp	Int symp	3	334	0.19	(0.11; 0.28)	9.65	0.01	0.27, *p* = 0.87	0.0%	0	12
	Dep symp	Dep symp	3	582	0.23	(−0.10; 0.51)	3.03	0.09	4.51, *p* = 0.10	55.6%	0.009	
	Dep symp	Ext symp	2	188	0.20	(−0.37; 0.67)	4.38	0.14	0.40, *p* = 0.53	0.0%	0	
2. Are current parental mental health symptoms associated with child mental health outcomes during the COVID-19 pandemic?	Psy-pathol	Psy-pathol	25	11,883	0.36	(0.31; 0.42)	12.26	<0.001	226.39, *p* < 0.001	89.4%	0.020	12,464
	Psy-pathol	Int symp	23	14,348	0.36	(0.30; 0.42)	10.90	<0.001	213.80, *p* < 0.001	89.7%	0.023	11,478
	Psy-pathol	Dep symp	8	6540	0.27	(0.16; 0.37)	5.93	<0.001	57.00, *p* < 0.001	87.7%	0.014	1076
	Psy-pathol	Anx symp	7	6018	0.27	(0.19; 0.34)	8.26	<0.001	25.15, *p* < 0.001	76.1%	0.006	823
	Psy-pathol	Ext symp	18	9103	0.32	(0.24; 0.38)	8.84	<0.001	151.49, *p* < 0.001	88.8%	0.021	5140
	Psy-pathol	Gen stress	5	1214	0.31	(−0.01; 0.57)	2.71	0.053	67.76, *p* < 0.001	94.1%	0.065	
	Dep symp	Psy-pathol	18	5692	0.39	(0.32; 0.44)	12.26	<0.001	100.62, *p* < 0.001	83.1%	0.015	4935
	Dep symp	Int symp	16	8357	0.28	(0.24; 0.33)	12.89	<0.001	35.01, *p* < 0.01	57.2%	0.004	2751
	Dep symp	Dep symp	11	14,958	0.27	(0.17; 0.37)	5.75	<0.001	196.72, *p* < 0.001	94.9%	0.023	2075
	Dep symp	Anx symp	8	6275	0.25	(0.21; 0.30)	11.96	<0.001	14.33, *p* < 0.05	51.1%	0.002	856
	Dep symp	Ext symp	13	4018	0.27	(0.22; 0.33)	10.62	<0.001	22.26, *p* = 0.03	46.1%	0.003	1194
	Dep symp	Gen stress	3	923	0.39	(−0.47; 0.87)	1.90	0.20	71.37, *p* < 0.001	97.2%	0.130	
	Anx symp	Psy-pathol	12	4110	0.31	(0.25; 0.38)	9.89	<0.001	43.49, *p* < 0.001	74.7%	0.008	1535
	Anx symp	Int symp	15	14,571	0.30	(0.23; 0.36)	9.85	<0.001	152.43, *p* < 0.001	90.8%	0.010	5098
	Anx symp	Dep symp	6	5613	0.22	(0.08; 0.35)	4.07	<0.01	41.34, *p* < 0.001	87.9%	0.015	481
	Anx symp	Anx symp	9	6549	0.26	(0.20; 0.33)	8.91	<0.001	36.32, *p* < 0.001	78.0%	0.006	1138
	Anx symp	Ext symp	9	3517	0.23	(0.17; 0.28)	8.80	<0.001	18.32, *p* = 0.02	56.3%	0.004	510
	Anx symp	Gen stress	3	923	0.33	(−0.41; 0.81)	1.91	0.20	34.72, *p* < 0.001	94.2%	0.088	
3. Are parents’ general stress levels associated with child mental health outcomes during the COVID-19 pandemic?	Gen stress	Psy-pathol	20	38,991	0.34	(0.29, 0.39)	13.24	<0.001	433.74, *p* < 0.001	95.6%	0.012	18,997
	Gen stress	Int symp	21	40,084	0.31	(0.27, 0.35)	14.37	<0.001	87.39, *p* < 0.001	77.1%	0.007	12,548
	Gen stress	Dep symp	10	5243	0.30	(0.21; 0.39)	7.39	<0.001	50.19, *p* < 0.001	82.1%	0.014	1510
	Gen stress	Anx symp	8	4913	0.32	(0.22; 0.41)	7.33	<0.001	29.68, *p* < 0.001	76.4%	0.012	1124
	Gen stress	Ext symp	17	36,844	0.31	(0.27, 0.35)	17.37	<0.001	161.94, *p* < 0.001	90.1%	0.004	10,637
	Gen stress	Gen stress	13	4110	0.29	(0.18, 0.39)	5.49	<0.001	139.59, *p* < 0.001	91.4%	0.033	1659
4. Is parenting stress associated with child mental health outcomes during the COVID-19 pandemic?	Par stress	Psy-pathol	9	4213	0.44	(0.33, 0.54)	8.32	<0.001	43.82, *p* < 0.001	81.7%	0.020	2387
	Par stress	Int symp	7	3307	0.46	(0.26, 0.63)	5.18	<0.01	95.89, *p* < 0.001	93.7%	0.059	1315
	Par stress	Dep symp	2	251	0.36	(−0.95; 0.99)	2.16	0.28	4.30, *p* = 0.04	76.7%	0.047	
	Par stress	Ext symp	8	4079	0.45	(0.39, 0.50)	16.37	<0.001	15.83, *p* = 0.03	55.8%	0.002	2354
	Par stress	Gen stress	2	1682	0.51	(−0.12, 0.85)	10.38	0.06	3.67, *p* = 0.06	72.8%	0.004	

anx symp, anxiety symptoms; dep symp, depressive symptoms; ext symp, externalizing symptoms; gen stress, general stress; *I*^2^, Higgins’ and Thompson’s *I*^2^ statistic; int symp, internalizing symptoms; *k*, number of studies included in meta-analysis; *o*, number of observations; *Q*, Cochran’s *Q* statistic; *p*, *p*-value; par stress, parenting stress; psy-pathol, psychopathology; *r*, Pearson’s product–moment correlation; *CI*, confidence interval; *Fail-safe N* for significant meta-analyses calculated using Rosenthal’s approach; *T*^2^, tau-squared statistic; *z*, *Z* value.

**Table 2 ijerph-20-04485-t002:** Summary of significant meta-analytic results on associations between parental variables prior to and during the COVID-19 pandemic and children’s mental health outcomes during the COVID-19 pandemic.

		**Child Variables** (all **during** the COVID-19 pandemic)					
		Psychopathology	Internalizing symptoms	Depressive symptoms	Anxiety symptoms	Externalizing symptoms	General stress
**Parental variables**	(1) **Prior to** the COVID-19 pandemic						
	Depressive symptoms		0.19				
	(2) **During** the COVID-19 pandemic						
	Psychopathology	0.36	0.36	0.27	0.27	0.32	
	Depressive symptoms	0.39	0.28	0.27	0.25	0.27	
	Anxiety symptoms	0.31	0.30	0.22	0.26	0.23	
	General stress	0.34	0.31	0.30	0.32	0.31	0.29
	Parenting stress	0.44	0.46			0.45	

Numbers are Pearson’s product–moment correlations *r*; all specified correlations were significant with *p* < 0.05.

## Data Availability

The data presented in this study are available on request from the corresponding author.

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
