# Peer review of "Mental Health Is a Family Affair—Systematic Review and Meta-Analysis on the Associations between Mental Health Problems in Parents and Children during the COVID-19 Pandemic"

_ijerph, 2023, doi:10.3390/ijerph20054485_

Round 1
Reviewer 1 Report
Please see the attached pdf

Reviewer 2 Report
This article has an interesting topic for clinical practice. However, it needs improvement. I will make some of them below:
The title is confusing, it is difficult to understand the scope of the study. I would suggest something like "Mental health is a family affair - implications of the pandemic for COVID-19 on parents and children"
This review has many research questions that could have been split into different reviews to be better explored - suggestion for future reviews.
In "Methods" the authors should present the type of review with a rationale for the choice by with appropriate references relating to the aim of the review.
Although the search strategy is in supplementary material, it would be important to present here with some systematization the inclusion and exclusion criteria such as type of study, language, access to full text or not, quality level of studies to be included ...
In "Risk of bias" the choice of instrument should be justified.
Across-sectional studies and longitudinal studies were included, but the assessment tool only applies across-sectional studies. How was the quality of longitudinal studies assessed?
When it is stated "In order to address the research questions, the following information was extracted from the included studies: author(s), year of publication, title, country, study design (e.g. online survey, cross-sectional), parent-/child-report, study period, study participants sample size, gender distribution, children's age), measurement of parental mental health problems, measurement of parents' general stress, measurement of parenting stress, measurement of children's mental health outcomes, association of parental mental health problems/stress and children's mental health outcomes. Before being a meta-analysis, it is a systematic review, so it is necessary to know and present this data.
The discussion is poor, it does not deepen the analysis of the findings. It is necessary to analyse the relevance of the findings in comparison to pre-pandemic studies and discuss what this study brought new to these previous results, looking for justifications for differences or different results than expected.
Suggestions for future studies should be made.
The Conclusion does not integrate the relationship between the results of the review and the conclusions it presents adequately, it should be improved. Regarding the impact of the pandemic on the mental health of parents and its relation to the mental health of children, it should be explicit here the significant findings and whether they are relevant compared to pre-pandemic studies.
They should check if the reference 12 is correct according to the journal's indications.
Reviewer 3 Report
This paper presents a systematic review and meta-analysis on an important topic. The subject is relevant, the aims are clear and you have chosen an appropriate research methodology, and I believe you will contribute to our understanding of this important aspect: parental effects of the 2 COVID-19 pandemic on children. Overall, I found the topic very interesting and the writing quite strong. Although I have some minor points the authors should consider, I would recommend accepting this piece of work for publication.
With that said, it is in the spirit of strengthening the manuscript that I offer the following questions/ comments/ recommendations:
- The title should specify that it is a systematic review and a meta-analysis.
- Please add the references in the first paragraph of the introduction.
- Table 1, some information is missing in notes
- Table 2, some information is missing in notes
Also, in discussion or conclusion you should add some thought about in what ways can future scholars build on your work? How might the findings of your study inform the work of clinicians/practitioners? How can clinicians use the findings of your study?
